# Characterization and Evaluation of Heat–Moisture-Modified Black and Red Rice Starch: Physicochemical, Microstructural, and Functional Properties

**DOI:** 10.3390/foods12234222

**Published:** 2023-11-22

**Authors:** Victor Herbert de Alcântara Ribeiro, Mario Eduardo Rangel Moreira Cavalcanti-Mata, Raphael Lucas Jacinto Almeida, Virgínia Mirtes de Alcântara Silva

**Affiliations:** 1Natural Resources Technology Center, Federal University of Campina Grande, Campina Grande 58019-900, PB, Brazil; virginia.m.alcantara@gmail.com; 2Department of Chemical Engineering, Federal University of Rio Grande do Norte, Natal 59056-000, RN, Brazil; raphaelqindustrial@gmail.com

**Keywords:** gelatinization, hydrothermal treatment, *Oryza sativa*

## Abstract

This study sought to evaluate starch from black and red rice modified by heat–moisture, investigating the extraction yield, starch and amylose content, color, and phenolic compounds. The water and oil absorption capacity, whole milk and zero lactose absorption index, syneresis index, and texture were also analyzed. Microstructural analysis included Fourier transform infrared spectroscopy, X-ray diffraction, and scanning electron microscopy. The heat–moisture treatment (HMT) reduced the extraction yield and the starch and amylose content, with native black rice starch having the highest values for these parameters. The modification also affected the color and phenolic compounds of the starch, making it darker and changing its appearance. The modification improved the absorption of water, oil, and milk, reducing syneresis and increasing stability during storage. The starch surface was altered, especially for modified black rice starch, with larger agglomerates. The type of starch also changed from A to Vh, with lower relative crystallinity. The textural properties of modified red rice starch were also significantly altered. The HMT proved to be a viable and economical option to modify the analyzed parameters, influencing the texture and physicochemical properties of pigmented rice starch, expanding its applications, and improving its stability during storage at temperatures above 100 °C.

## 1. Introduction

The National Supply Company (CONAB) (Brasília, Brazil) predicts that Brazil’s rice harvest for the 2022/23 period will be 7.2% smaller than the previous harvest, estimated at 10 million tons [1]. Brazil is among the 10 largest cereal producers in the world and is the largest rice producer on the American continent, with average annual production of 12.3 million tons, according to CONAB, [2]. Rice, consisting primarily of starch, protein, and lipids, serves as a crucial staple food for the majority of the world’s population [3].

The grain of rice is constituted mainly by starch, which can reach up to 90% of the grain. In this way, rice becomes an option in the industry for the production of various products that have starch as an important constituent [4]. Starch is a natural biopolymer found in vegetables, roots, tubers, cereals, legumes, and fruits. To improve the functional, chemical, and physical properties of native starch in order to improve its applicability in industry, polymer modifications are made by chemical, physical, enzymatic, or combined means (double modification) [5]. The polysaccharide has limited functionality in its native form, which can affect its applications [6]

Generally, starch chain structures can be altered through biological, thermal, non-thermal, hydrothermal, and chemical methods to achieve the desired technological, functional, and nutritional properties [7]. Physical methods are more practical, safe, and environmentally friendly for starch modification, where the purpose of physical modification is to alter the internal structure through the gelatinization of native starch granules. The resulting structural changes in starch due to heat-induced gelatinization have stabilizing effects, including improved solubility, stability under different conditions, viscosity and texture control, resistance to retrogradation, transparency, thermal stability, and a reduction in syneresis [8,9].

Black rice stands out among pigmented rice varieties because it has good sensory characteristics and high nutritional value. It has higher content of amino acids, organic acids, fatty acid methyl esters, and free fatty acids, and, being gluten-free, it becomes an excellent raw material for the development of products for patients with celiac disease or gluten sensitivity. Some of the polyphenols found in black rice grains include anthocyanins, proanthocyanidins, ferulic acid, ellagic acid, and quercetin [10].

Ito & Lacerda, [11] found that red rice cultivated in Brazil, a special type of rice whose main differential is the reddish color of the grain, is largely produced through family farming, mainly in the semi-arid region of Paraíba, Rio Grande do Norte, Pernambuco, and Ceará. While Lima et al. [12] found that due to the distinct flavor, color, aroma, cooking, and functional characteristics, special types of rice, such as red rice, are consumed by specific market niches, and even these characteristics have been important to increase the value of these types of rice.

Heat–moisture treatment (HMT) promotes physicochemical modifications in the granule without changing its molecular composition, where it is carried out at a temperature above the gelatinization temperature, with an insufficient amount of water content to gelatinize the biopolymer [13]. HMT can also lead to the formation of resistant starch (undigested in the small intestine) upon cooling, where part of the starch recrystallizes into a more organized form known as resistant starch type 3 (RS3), which is beneficial in promoting intestinal health and aiding in blood glucose level regulation [14]. HMT can modify the structures of starch granules in rice by adjusting their layers, helical structures, and molecular organization. Initially, lipids and amylose coexist independently within the starch granules. However, during HMT with high moisture content, the granules gelatinize and expand sufficiently to allow the extensive crystallization or organization of molecules, resulting in the formation of an amylose–lipid complex [15,16].

The most recent applications of HMT-modified starches are ensuring the quality of 3D-printed wheat starch gels [17], the construction of bioactive delivery systems and Pickering emulsions [18], the development of nanocomposite films [19], and application in starch noodles and potato [20]). The main problem is the complexity of some of these technologies and the high cost of their implementation, mainly caused by the high price of the gelling agent, commonly agar. The cause of the high price of the latter input is its overexploitation, a consequence of high demand, which has made it increasingly difficult to obtain, raising prices [21]. Therefore, this work aims to characterize and evaluate the physical-chemical, microstructural, and functional properties of black and red rice starch modified by heat–moisture treatment (HMT).

## 2. Materials and Methods

### 2.1. Materials

Grains of black and red rice (Timbaúba^®^, Timbaúba, Brazil) were purchased from local companies in the city of Campina Grande, Paraíba, where they were taken to the Drying Laboratory (UAEALI) of the Federal University of Campina Grande (UFCG) for the removal of impurities and manual selection. Until the rice starch extraction, the samples were stored at room temperature (25 ± 2 °C).

To carry out this research, the following materials were acquired: lactic acid (FCC85 food-grade solution, Purac, Gorinchem, The Netherlands), whole milk and zero lactose (Betânia^®^, Parnamirim, Brazil), sunflower oil (Liza^®^, Mairinque, Brazil), ethyl alcohol, 3,5-dinitro salicylic acid (NEON^®^, São Paulo, Brazil), sodium acetate (Synth^®^, Diadema, Brazil), acetic acid (Anidrol^®^, Diadema, Brazil), resublimated iodine (Innova^®^, Recife, Brazil), soluble starch (Synth^®^, Diadema, Brazil), pectin (Grindsted^®^Pectin YF 310, Danisco Mexicana, DuPont, Lerma, Mexico), sodium metabisulfite and dimethylsulfoxide—DMSO (Dinâmica^®^, São Paulo, Brazil), and potassium iodide (Synth^®^, Diadema, Brazil).

### 2.2. Starch Extraction

To perform starch extraction, the method initially described by Bento et al. [22] and later modified by Almeida et al. [23] was used. First, black rice and red rice were immersed, separately, in a sodium metabisulfite solution (0.5%) with a ratio of 1:2 (*w*/*v*) for 72 h at a temperature of 7 °C. Subsequently, it was rinsed in running water for 3 min, thus proceeding to the crushing stage, which occurred by adding distilled water in the proportion of 1:2 (*w*/*v*) with the aid of an industrial blender (KD Eletro, model Lar-22, Itajobi, Brazil) over 5 min. After this process, filtering through an organza mesh (0.3 mm) was performed to obtain the starch suspensions. Again, the residues were ground and filtered four times to increase the extraction yield. From this point on, the starch suspension was passed through a 32-mesh sieve and was decanted five times every 12 h in a domestic refrigerator (Electrolux, model RE31, São José dos Pinhais, Brazil) at 7 °C to avoid the occurrence of enzymatic or fermentative action during the sedimentation process. Vacuum filtration was used to remove the remaining water and thus optimize the process.

### 2.3. Heat–Moisture Treatment (HMT)

The final water content of 20% was achieved by adding the black and red rice starches (3 g) to distilled water in conical flasks. After this, these flasks were sealed and heated at a temperature of 110 °C for 3 h (HMT3) with the aid of an electric oven (Semp Easy, model FO3015PR2, São Paulo, Brazil) with dimensions of 25 × 41.5 × 32.2 cm [24]. Subsequently, the samples were dried in an oven with air circulation, at a temperature of 50 °C and fixed air velocity of 1.0 m s^−1^, followed by grinding with a knife mill (Biofoco, model BF2 MMH 27000, São José dos Campos, Brazil) [25].

### 2.4. Determination of Starch and Amylose Content

Starch determination was performed using a methodology adapted from Amaral et al. [26]. First, 10.0 mg of the sample and four extractions with 500.0 μL of 80% ethanol at 80 °C for 20 min were used, aiming to remove sugars, pigments, phenols, and other soluble substances and to determine the starch dosages. Subsequently, 1.0 mL (30.0 U g^−1^) of α-amylase diluted in 0.016 M sodium acetate buffer pH 6.0 was added. For 30 min, the samples were incubated at 75 °C and, after this, they were cooled to 50 °C and then 1.0 mL of a solution comprising 70.0 U g^−1^ of amyloglucosidase in 100 mM (pH 4.5) sodium acetate buffer was added. After this, the samples were incubated for 30 min at 50 °C and centrifuged (10,000× *g* for 2 min). Afterward, the starch was measured by quantifying the glucose released in the process according to the methodology proposed by Miller [27].

In agreement with the descriptions by Nadiha et al. [28] and McGrance et al. [29], the amylose content was analyzed using 10 g of starch together with 2 mL of DMSO, which was then heated at a temperature of 85 °C for 15 min. Deionized water was added to increase the volume of the solution to 25 mL, and then the pipetting process was conducted using 1 mL of the starch solution in a volumetric flask with a capacity of 50 mL, followed by the addition of 5 mL of iodine solution to reach 50 mL. The iodine reagent was prepared by dissolving potassium iodide (20 g) and resublimated iodine (2 g) in 100 mL of water. The determination of the absorbance of the sample took place at a wavelength of 620 nm. Both parameters were expressed in g 100 g^−1^ of starch on a dry basis.

### 2.5. Color Determination

The CIELAB color scale CIELAB (L*, a*, b*) was chosen for the measurement of starch color. The registration of the color values was based on the average of the values of five photographs, where the parameters included L* = brightness (0 = black, 100 = white), a* (−a* = green, +a* = red), and b * (−b* = blue, +b* = yellow).

The starch color was determined by measuring the coordinates L* (lightness), a* (redness), and b* (yellowness), using a digital colorimeter (X). For the reading, the following conditions were set: illuminant C, viewing angle 8°, standard observer angle 10°, specular included, according to the specifications of the Commission Internationale de L’éclairage [30]. Before taking the readings, the instrument was calibrated on a white ceramic plate (Illuminant C: Y = 92.84 X = 0.3136, y = 0.3201).

### 2.6. Extraction Yield

The extraction yield was established from the ratio between the starch weight and the seed weight, both expressed on a dry basis using Equation (1):(1)$$Y=\frac{W_{db}}{W_{t}}\times 100\;$$
*Y* is equivalent to the extraction yield, *W_db_* to the weight of starch in the grains on a dry basis (g), and *W_t_* to the weight of the grains (g).

### 2.7. Determination of Phenolic Compounds by High-Performance Liquid Chromatography (HPLC)

Phenolic compounds were obtained by using 0.1 g of starch diluted in 5 mL of acetic acid and ethanol solution (1.5:1). There was filtration through a 0.22 µm membrane followed by HPLC analysis using the SPD-10AV-VP (UV-VIS detector, Shimadzu, Kyoto, Japan). Separation was performed on a Shim-Pack CLC-ODS C18 column (Kyoto, Japan) (4.6 mm × 15 cm) at a temperature of 25 °C, with a mobile phase draining at a flow rate of 0.6 mL min^−1^ for 17 min. As for the composition of the mobile phase, it consisted of 1% acetic acid (*v*/*v*), ethanol (1.5:1) (phase A), and HPLC-grade acetonitrile (phase B). A volume of 20 μL of the samples was applied to the system, and the run occurred using the following gradients: from 100% to 70% of phase A (0–5 min), followed by 70% to 30% of phase A (5–10 min) and 30% to 0% of Phase A (10–12 min) [23]. Finally, the run ended after 5 min with 0% phase A, totaling 32 min of running. The parameters determined were gallic acid (GA), proanthocyanidins (P), and quercetin (Q), as shown in Equations (2)–(4) and expressed in (mg L^−1^). Next, identification was carried out under the same treatment conditions as the respective standards, and the quantification of the phenolic compounds was based on the retention time (GA: 3.41 min; P: 10.4 min; and Q: 12.75 min) and peak area (A).
GA = 3.10 × 10^−5^ × A(2)
P = 2.69 × 10^−5^ × A(3)
Q = 4.67 × 10^−5^ × A(4)

### 2.8. Water Absorption Capacity

The water absorption capacity was determined from the method Beuchat, [31], in which 1 g of starch was used with 10 mL of distilled water. Homogenization occurred for 30 min and the sample was kept in suspension for another 30 min. Subsequently, each sample went through the centrifugation process (VitchLab, model K14–5000 M, Araras, Brazil) for 15 min at 3000× *g*. Absorption was calculated according to Equation (5):(5)$$CAA=(1-\frac{Ms}{Ms+Mw-Ew})\times 100$$
*CAA* represents the water absorption capacity expressed in (g 100 g^−1^), *Ms* is the initial mass of starch on a dry basis (g), *Mw* is the mass of water (g), and *Ew* is the mass of excess water (g).

### 2.9. Oil Absorption Capacity

A combination of 1 g of rice starch and 10 mL of sunflower oil was created in centrifuge tubes. Samples were kept at 24 °C, manually shaken for 5 min, and then centrifuged (SOLAB, model SL706, Piracicaba, Brazil) at 15,000× *g* for 15 min. After this process, excess oil was removed and weighed. The calculation of the amount of oil retained per gram of sample was performed according to Equation (6) [31]:(6)$$CAO=(1-\frac{Ms}{Ms+Mo-Eo})\times 100$$

Here, *CAO* corresponds to the absorption capacity of oil expressed in (g. 100 g^−1^), *Ms* is the initial mass of starch on a dry basis (g), *Mo* is the mass of oil (g), and *Eo* is equivalent to the mass of the oil excess (g).

### 2.10. Whole Milk and Zero Lactose Absorption Index (IAL)

Starch (2.5 g) was added to 30 mL of milk at a temperature of 25 ± 2 °C for 30 min and then centrifuged at 3000× *g* for 15 min (SOLAB, model SL706, Piracicaba, Brazil). Soon after centrifugation, the supernatant was transferred to a Petri dish of known mass, followed by the weighing process. The milk absorption index, which corresponds to the mass of the gel obtained after removing the supernatant, was acquired by Equation (7) [32]:(7)$$IAL=(1-\frac{Ms}{Ms+Mrg-Msu})\times 100$$
where *IAL* is the milk absorption index (g. 100 g^−1^), *Ms* corresponds to the mass of starch on a dry basis (g), *Mrg* to the mass of the centrifugation residue (g), and *Msu* to the supernatant mass (g).

### 2.11. Syneresis Index

The determination of the syneresis index occurred according to the methodology proposed by Farnsworth et al. [33]. Starch pastes were prepared in a proportion of 10% (starch/water) (*w*/*v*) at a temperature of 80 °C for 12 min, controlling the temperature with a thermostatic bath (LaborgLas, model ALPHA A12, Belenzinho, Brazil). The samples spent 72 h stored at 7 °C and, subsequently, there was 15× *g* centrifugation (VitchLab, model K14-5000 M, Araras, Brazil) at 3000× *g* for 10 min. The analysis was performed in triplicate and the syneresis index was calculated according to Equation (8).
(8)$$Syneresis(\%)=\frac{m_{f}}{m_{i}}\times 100$$
where *m_f_* corresponds to the mass of water separated from the gel after centrifugation (g) and *m_i_* is the initial mass of the gel (g).

### 2.12. Fourier Transform Infrared Spectroscopy (FT-IR)

To obtain FT-IR spectra in the region of 650–4500 cm^−1^, the Cary 630 equipment (Agilent Technologies, Santa Clara, CA, USA) was used, with a resolution of 4.0 cm^−1^ and 32 scans [34]. To deconvolve the FT-IR peaks, the Gaussian function was used and the ratio between the bands at 1047 cm^−1^, 1022 cm^−1^, and 995 cm^−1^ was performed by recording the absorbance height to characterize the molecular order [35].

### 2.13. X-ray Diffraction (XRD)

The process of determining the X-ray crystallographic patterns of the samples was carried out with the aid of an X-ray diffractometer (Shimadzu, model XRD-7000, Kyoto, Japan) equipped with Cu-Ka radiation (λ = 1.5406 Å) at 80 mA and 40 kV. A scan was performed at a 0.02° step, with a counting time of 2 s from 10° to 40° at room temperature. Intending to convolve the diffractogram and reveal the amorphous area, the Gaussian response function was used. Soon after, the crystallinity of the samples was established by the area method using Equation (9) [36]:(9)$$IC(\%)=\frac{A_{c}}{A_{t}}\times 100$$
where *IC* is the crystallinity index, *A_c_* is the crystalline area below the peaks, and *A_t_* is the total area.

### 2.14. Scanning Electron Micrographs (SEM)

The samples were analyzed before and after hydrothermal modification by scanning electron micrographs (Shimadzu, Superscan SSX-550, Kyoto, Japan). Then, they were evenly dispersed on the sample table, to which a conductive double-adhesive carbon tape was attached and covered with a thin layer of gold. Then, 1000× magnification was used with an acceleration potential of 10 kV and the measurement of the granule size was performed with the help of the Image J 1.0 software http://rsbweb.nih.gov/ij/ (accessed on 7 September 2023).

### 2.15. Texture Profile

The preparation of red rice starch pastes occurred with the proportion of (1:10) starch–water that was heated at a temperature of 80 ± 2 °C until pastes were formed in approximately 30 min. The samples were stored at a temperature of 8 °C in a refrigerator (Brastemp, model BRM44, São Bernardo do Campo, Brazil) for 24 h; they were then placed at room temperature (25 ± 2 °C) and examined using the TPA method, which associates textural parameters with sensory analysis [25]. To determine the texture of the starch pastes, a tetrameter (Stable Micro Systems, model TA-XT PLUS, Godalming, UK) was used, with a sample holder of 50 mm in diameter and 40 mm in height. The starches were compressed twice at 40% at a speed of 2 mm/s. The calculation of cohesiveness, gumminess, firmness, and elasticity values was performed from the force-per-time plots obtained [37].

### 2.16. Statistical Analysis

The data were expressed as mean ± standard error (M ± SD) calculated in triplicate for each analysis, where they were subjected to ANOVA using the Tukey test (*p* < 0.05), with data processing using Statistica^®^ v.7 (Statsoft, Tulsa, OK, USA).

## 3. Results and Discussion

### 3.1. Characterization of Native and Modified Black and Red Rice Starch by Heat–Moisture (HMT)

The physical and chemical evaluation, related to color parameters, extraction yield, starch, and amylose content, can be seen in Table 1.

Regarding the starch extraction yield (Table 1), it was observed that there were significant differences between the formulations. Black rice starch (AP) showed the highest extraction yield (52.06%), followed by red rice starch (AV) (49.11%). Furthermore, Moura et al. [38] found a yield for native starch extraction from black rice of 50.03% and for red rice of 67.84%. These values were close to the AP value but higher than AV. Modified black rice starch (AP_HMT_) and modified red rice starch (AV_HMT_) had lower yields (44.31% and 41.07%, respectively). The starch extraction yield was affected by HMT, as, in the steam processes, the high moisture content and high condensation temperature of the steam on the starch surface resulted in partial gelatinization, making extraction more difficult [39]. Almeida et al. [23] observed a 17.90% reduction in the extraction yield of red rice starch after being subjected to autoclaving. Meanwhile, Almeida et al. [25] found a smaller decrease (3.25%) for HMT-modified quinoa starch.

For both starches, a significant reduction in starch content was noted after HMT. HMT can also cause the breaking of glycosidic bonds in starch, leading to the fragmentation of starch molecules into smaller parts. This can contribute to a reduction in starch content [40].

According to Ramli et al. [41] and Kumar et al. [42], the reduction in amylose content in rice, as observed in this study, can be attributed to the application of HMT. HMT induces the partial gelatinization of starch, leading to the disintegration of its crystalline structures and resulting in a more amorphous structure. Consequently, amylose and amylopectin separate and disperse in water, which can result in the loss of amylose [43].

These results highlight the predominance of amylopectin in the black and red rice starches used in this research, as the relatively high proportion of amylopectin in relation to amylose can influence the functional properties of these starches, such as the gelatinization capacity, texture stability, and thickening [44,45].

We then considered the color parameters (L (brightness), a* (+a) red, (−a) green, b* (+b) yellow, and (−b) blue) among the starch formulations. Chroma a* showed a difference and chroma b* showed the same behavior under background conditions for AP and AP_HMT_ samples, indicating the independence of shades in the appearance of starches. The color analysis results revealed significant differences between black (AP) and red (AV) rice starch formulations. Regarding brightness (L*), AP starch obtained an average value of 62.71, indicating a darker hue, while AV starch had an average value of 71.16, indicating a lighter hue. As for the chroma a*, AP starch obtained an average value of 4.92, while AV starch obtained an average value of −0.78, indicating a redder hue for AP starch. On the other hand, chroma b* showed mean values of 6.52 for AP starch and 10.16 for AV starch, indicating differences in the yellow/blue hue between the formulations.

The luminosity of the products showed a difference between their values that may have resulted from the coloring of the starches. In Figure 1, it is possible to observe the different color tones of the starches obtained from rice with and without treatment. If the color is darker, it can directly influence parameters such as luminosity in relation to the formulations derived from the raw material used.

These results confirm the differences in the color properties of black and red rice starches, highlighting their potential applications in various industries that require specific color characteristics in their products. These differences can be attributed to the chemical composition and intrinsic characteristics of the black and red rice grains, as well as the processing conditions used to obtain the starches. According to Elias et al. [46], the prolonged application of moist heat and pressure treatment can lead to a higher occurrence of damaged starch and the more pronounced darkening of the color. The different values of luminosity and the a* and b* chromas are related to the conditions used to obtain the samples [47]. Higher values for chroma a* (red) indicate a darker color for the sample. As for chroma b*, a tendency to yellow results from the presence of beta-carotene [48]. The change in color in starches after heat–moisture modification can occur due to various reasons, including Maillard reactions, the degradation of natural pigments, and alterations in the structure and composition of starch [49].

### 3.2. Determination of Phenolic Compounds by HPLC

Native and HMT-modified rice varieties showed differences in all phenolic compounds found: gallic acid, proanthocyanidins, and quercetin (Table 2). It was observed that the native starches of black and red rice after processing with HMT showed a reduction in the amount of these compounds. The analysis of the phenolic compounds presents in red and black rice starches, both in native form and in HMT-modified form, revealed significant differences between the formulations.

Regarding gallic acid, the native starches of black (AP) and red (AV) rice showed higher values, with 2.49 mg of gallic acid L^−1^ and 0.91 mg of gallic acid L^−1^, respectively. On the other hand, HMT-modified starches (AP_HMT_ and AV_HMT_) showed lower values, with 1.52 mg of gallic acid L^−1^ and 0.63 mg of gallic acid L^−1^, respectively. These results indicate that HMT processing resulted in a significant reduction in the amount of gallic acid in the starches. The values of gallic acid obtained in this study are in agreement with previous studies that also reported similar levels of this compound in rice. Mira et al. [50] obtained values of gallic acid involving rice similar to those of this study of 0.69–0.98 mg of gallic acid L^−1^. According to Tyagi et al. [51], black and red rice have greater antioxidant activity in their compositions due to the levels of phenolic compounds present. The presence of these compounds can guarantee the development of functional products to combat free radicals as protection against oxidative damage at the cellular level.

In the case of proanthocyanidins, black rice starches (AP and AP_HMT_) showed lower values, with 0.71 mg of catechin L^−1^ and 0.48 mg of catechin L^−1^, respectively. Red rice starches (AV and AV_HMT_) showed higher values, with 1.37 mg of catechin L^−1^ and 1.02 mg of catechin L^−1^, respectively. These results demonstrate significant differences in proanthocyanidins between different types of starch, indicating that HMT processing can affect the presence of this compound. The proanthocyanidin values became consistent when comparing both starches (black and red). According to Mackon et al. [52], the greater the amount of proanthocyanidins found in a vegetable raw material, such as rice, the redder it becomes.

In the case of quercetin, the native starches of black (AP) and red (AV) rice showed higher values, with 2.96 mg of quercetin L^−1^ and 1.95 mg of quercetin L^−1^, respectively. HMT-modified starches (AP_HMT_ and AV_HMT_) showed lower values, with 2.58 mg of quercetin L^−1^ and 1.20 mg of quercetin L^−1^, respectively. These results show that HMT processing also resulted in a reduction in the amount of quercetin in the starches.

These data indicate that modification by HMT affected the presence of phenolic compounds in black and red rice starches. The reduction in the levels of gallic acid, proanthocyanidins, and quercetin in modified starches may have nutritional and functional implications since these compounds have bioactive properties. It is important to consider these changes when using black and red rice starches in food and industrial applications, taking into account the properties of phenolic compounds and their potential effects. Furthermore, the reduction in the amount of quercetin in starches modified by HMT is in agreement with studies that show a decrease in this compound in vegetables subjected to temperature increase processes, according to the research by Harnly et al. [53], who claim that the amount of quercetin can be reduced in vegetables with the application of processes that involve increasing the temperature, which could obtain values between 0.000 and 0.600 mg of quercetin per liter after cooking, corroborating the results for the starches modified by the HMT process compared to native black and red rice starch.

### 3.3. Functional Analysis of Starches Obtained from Black and Red Rice

Among the native starches, red rice starch showed the highest water absorption capacity (70.01 g 100 g^−1^) (Table 3). This was confirmed by Moura et al. [54] for the native starches of black and red rice. When observing the starches modified by HMT, both showed similar values, with no significant differences (*p* < 0.05), and there was also an increase compared to the native starches, with AP_HTM_ showing a 9.87 g 100 g^−1^ increase. The application of starch modification techniques, such as HMT, can interfere with characteristics related to the solubility index and functional properties involving water absorption, gelation, and pasting [55].

HMT-modified black rice starch (AP_HMT_) has the highest oil absorption (72.12 g 100 g^−1^) among the formulations. Native red rice starch (AV) shows the lowest oil absorption (58.52 g 100 g^−1^), while HMT-modified black rice starch (AV_HMT_) and HMT-modified red rice starch (AV_HMT_) have intermediate values of 65.96 and 65.96 g 100 g^−1^, respectively. Almeida et al. [56] noted that the binding capacity to oil and milk is more efficient when native red rice starch is hydrothermally treated.

Modified red rice starch (AV_HMT_) has the highest absorption from lactose-free milk (74.24 g 100 g^−1^), followed by modified black rice starch (AP_HMT_) (73.58 g 100 g^−1^). The opposite is verified for the absorption of whole milk, where AP_HMT_ presents the highest value (73.93 g 100 g^−1^). When evaluating enzymatically modified native red rice starch, Almeida et al. [25] reported an oil absorption value of 47 g 100 g^−1^ and a whole milk absorption value of 50 g 100 g^−1^.

It is observed that AV starch, in relation to AP, AV_HMT_, and AP_HMT_, has a higher syneresis index over five cycles, around 40% in relation to the others (Figure 2). It is related to the reorganization of the starch structure during the loss of water after the gelatinization process and the formation of amylopectin crystals resulting from this loss. Functional aspects involving starch syneresis become important in evaluating retrogradation processes, in which starch releases water and contracts, forming a more dense and cohesive mass. Syneresis is more pronounced in starches with high amylose content and long chains of amylopectin.

The result demonstrates that native and HMT-modified black and red rice starch is more prone to retrogradation, due to the high rate of water loss, mainly from the AV formulation of around 40%. Syneresis occurs mainly between amylose formations, which, when above 100 glucose units, tend to retrograde, and long chains of amylopectin [57].

According to Damodaran and Parkin [58], the higher the concentration of amylose in the starch, the greater the propensity for the occurrence of the syneresis phenomenon. However, the correlation between the amount of starch/amylopectin, the rearrangement of the two molecules, and the presence of other compounds, among others, can promote water loss (expulsion of water—syneresis). The lowest syneresis was obtained by the formulation of modified black rice starch (approximately 30%). This characteristic gives the product greater potential for application in food products that require storage under refrigeration, causing them to lose less free water to the environment.

### 3.4. Scanning Electron Microscopy (SEM)

Regarding the morphological part of the AP (Figure 3A) and AV (Figure 3C) starch granules, they have predominantly oval, truncated ellipsoid, and/or hemispherical shapes. However, after HMT treatment, AP and AV (Figure 3B,D) demonstrate some granule dispersibility and the formation of bulky parts. These characteristics indicate that the HMT modification process leads to the greater agglomeration of the starch granules, resulting in a more compact structure. It is interesting to note that structural changes are more evident in modified black rice starch (AP_HMT_), possibly due to the presence of a higher amount of amylose in its composition compared to modified red rice (AV_HMT_).

The starch present in rice can be spongy and have undefined shapes. It is also possible to observe that the black and red rice starch granules present regular formation of their shapes [59]. Ziegler et al. [60] discovered that the starch granules obtained from brown, black, and red rice shared the same geometry, and they did not detect significant differences based on the coloration of the grain pericarp. Minor variations were observed in the surface appearance. It is an important feature to be verified by SEM that can have an integrative effect on the reactivity of the starch when modified, as well as on its technological and biological characteristics. This information is relevant to understanding the structural and functional characteristics of starch and its potential application in different food products.

Regarding the average diameter (Table 4) of the starch particles, the AP and AV samples showed smaller values (1.81–6.12 µm and 1.55–6.28 µm, respectively), indicating particles of smaller size. On the other hand, samples modified by HMT, AP_HMT_, and AV_HMT_ showed larger mean diameters (57.63–588.23 µm and 29.41–352.94 µm, respectively). These results suggest that the HMT modification process resulted in larger starch particles. According to Shi et al. [61], the average diameter of the starch granule present in rice is up to 5 μm depending on the type of rice. It becomes evident that the values found for the starch diameters for AV and AP are consistent with the literature. The relative crystallinity allows the observation of new bonds between the glucose molecules that form amylose and amylopectin, which may result in greater crystallinity [62]. The difference found between the results of native and modified starches indicates that when applying HMT, there may have been an increase in the number of crystallines [63].

### 3.5. X-ray Diffraction Analysis

It is evident that the XRD profiles of the rice starch samples exhibit type A diagrams, commonly found in cereal starches (Figure 4), except for the AP_HTM_, which is of the Vh type, due to the flattening of the peaks at 17° and 18° and the presence of the peak at 19.37°. According to Fonseca et al. [64] diffractograms outline type A spectra, demonstrating peaks of refraction intensity at angles at 2θ at 15°, 17°, 18°, and 23° that are evident in the starch found in cereals, whose refractions can also be observed in Figure 4. Yang et al. [65] indicate that hydrothermal effects promote intra- and intermolecular rupture in starch granules, leading to starch changes and a reduction in molecular packing.

It is possible to observe that the intensity between the bands present in the diffractograms in Figure 4 differs depending on the starch studied (native or modified). These differences are evidenced in bands around 15° and 20° in both black and red rice starches, respectively. Modified starches AP_HMT_ and AV_HMT_ show greater differences in peak intensity within the range of 20°, with AP_HMT_ representing a greater absorption spectrum compared to the other samples.

Note that the native starch granules have smaller diameters when compared to their modified structures, resulting from their gelatinization, with AP_HMT_ having greater dispersibility of these granules. Relating the relative crystallinity (RC) data (Table 4), it is observed that AP and AV do not differ, indicating that the arrangement of compounds such as amylose and amylopectin is similar in the organization of the starch structure. However, we obtained different RC results for modified starches, resulting from the reorganization/restructuring of the starch chain after the HMT process.

Regarding the relative crystallinity, it is observed that the AP and AV samples presented higher values (24.03% and 23.88%, respectively), indicating a greater proportion of crystalline regions in the starch. On the other hand, the samples modified by HMT, AP_HMT_, and AV_HMT_ presented lower relative crystallinity values (15.08% and 21.69%, respectively), suggesting that the HMT modification process may result in a decrease in starch crystallinity.

### 3.6. Fourier Transform Infrared Spectroscopy (FT-IR)

There are bands in the region of 3300 cm^−1^, related to the stretching of O–H groups, bands referring to the region of 2927 cm^−1^ that involve C–H, and those at 1643 cm^−1^ and 1327 cm^−1^ that may be associated with water bound in starch and the vibrations of CH_2_ groups, respectively (Figure 5A) [66].

In Figure 5B, one can see a band in the region of 1150 cm^−1^ related to the α-1,4, C–O–C glycosidic bond, while 1076 cm^−1^ and 1047 cm^−1^ are related to elongations between C–O and C–C [67]. Band patterns involving regions between 1022 cm^−1^ and 995 cm^−1^ are associated with the asymmetric vibrational motion of H–O–H groups [68]. It is possible to notice that the infrared spectra formed for AP_HMT_ have lower absorption between the regions of bands observed in relation to the other starches, AP, AV, and AV_HMT_.

Regarding the values obtained for IR (1022/1047 cm^−1^), the starches modified by HMT in black and red rice did not differ statistically, regarding AP and AV, demonstrating that the functional groups and the chemical bonds that constitute the modified rice starch remained constant. However, when relating wavelengths at 995/1022 cm^−1^, both native and modified starches differed from each other (Table 4). When comparing the IR (1022/1047) and IR (995/1022 cm^−1^) values, differences between the samples could be noticed. For IR (1022/1047 cm^−1^), AP and AV samples showed intermediate values of 1.39 and 1.45 cm^−1^, respectively, while AP_HMT_ and AV_HMT_ showed slightly lower values of 1.34 and 1.33, respectively. As for IR (995/1022 cm^−1^), the AP_HMT_ (1.08 cm^−1^) and AV_HMT_ (1.28 cm^−1^) samples showed lower values, while AP and AV showed intermediate values of 1.39 and 1.45 cm^−1^, respectively. These results indicate that the modification process by HMT can lead to changes in the molecular structure of starch.

The IR at wavelengths 1022/1047 cm^−1^ and 995/1022 cm^−1^ allows the observation of regions of greater starch opacity caused by the formation of crystals formed in the retrogradation process that directly affect its IR [24]. Furthermore, according to Jung et al. [69], when the starch goes through the gelatinization process and undergoes retrogradation, the water loss process causes the hydrogen bonds to become stronger in an attempt to reassociate the carbohydrate structure.

### 3.7. Analysis of Texture Parameters

Table 5 presents a detailed comparison of the texture parameters between native and modified starches.

Texture is an important parameter when it comes to product acceptance by consumers and reflects the mechanical resistance in relation to chewing and the sensory evaluation of starch pastes [70]. After modification, the degree of reassociation of amylose molecules is proportional to the intensity with which starch undergoes the gelatinization process, and this directly influences its texture parameters [64].

It can be seen that there were no significant differences in adhesiveness for the AP and AP_HMT_ formulations. However, for the remaining parameters—firmness, gumminess, and cohesiveness—statistical differences were evident, indicating that HMT modification has a direct and significant influence on starch texture. It is observed in Table 5 that among the texture parameters of the analyzed native and HMT-modified starches, only the adhesiveness for the AP and AP_HMT_ formulations did not differ. However, for the other parameters of firmness, gumminess, and cohesiveness, the presented formulations showed statistical differences, showing that the HMT treatment directly influenced the texture of the starch. In the firmness parameter, a decreasing tendency was observed as the modification process by HMT occurred. The native AP starch had the highest firmness (0.52 N), while the modified AV_HMT_ starch had the lowest firmness (0.34 N). This indicates that the modification process can lead to a softer and less firm texture.

The analysis of texture parameters in native and HMT-modified starches is essential to understanding how structural changes can influence the physical characteristics of these polymers. Modification by HMT can change properties such as adhesiveness, firmness, gumminess, and cohesiveness, affecting the behavior of starch in various applications, including food and materials [71]. These results corroborate those shown in Figure 5, in which the AP_HMT_ formulation shows a lower rate of syneresis.

Gumminess, which is related to the sensation of chewing and the release of flavor during consumption, also showed a decrease in values as the modification by HMT occurred. The native AP starch had the highest gumminess (0.42 N), while the modified AV_HMT_ starch had the lowest gumminess (0.22 N). According to Silva et al. [72], the gels formed by black and red rice starches require little energy to be swallowed, making them easily chewable, which can be a desirable feature in food products, providing a pleasant sensory experience during ingestion. The low gumminess resulting from HMT can generate a soft starch gel that can disintegrate easily during chewing [23].

Cohesiveness, which is related to the ability of starch to hold together during mastication, also shows a tendency to decrease in values with modification by HMT. The native AP starch had the highest cohesiveness (0.74 N), while the modified AV_HMT_ starch had the lowest cohesiveness (0.52 N). The values demonstrated in this study for the texture parameters indicate that black and red rice starch form harder gels from the long chains of amylopectin, as stated by Ramos et al. [59]. Cohesiveness is related to the ability of the starch to remain together during mastication, and the values obtained suggest that the gels formed by the black and red rice starches have a good capacity to maintain their cohesive structure during the mastication process [73]. High cohesiveness is viewed as a reliable indicator for the categorization of gels since it reflects a sample’s capacity to retain its initial form following initial compression [74].

Adhesiveness, which is related to the ability of starch to adhere to teeth and oral tissues, shows a decrease in values with modification by HMT. The native AP starch had the highest adhesiveness (0.49 ± 0.02 N.m), while the modified AV_HMT_ starch had the lowest adhesiveness (0.35 ± 0.01 N.m). High adhesiveness can bring an undesirable rubbery mouthfeel [56].

## 4. Conclusions

Native and modified black and red rice starches showed different characteristics when compared according to the studied parameters. The hydrothermal modification by heat–moisture (HMT) decreased the extraction yield and the starch and amylose content, where the native black rice starch (AP) presented the highest values of 52.06% and 82.06 g 100 g^−1^ for the first two parameters, respectively. As for the amylose content, native red rice starch (AV) stood out (15.46 g 100 g^−1^).

The HMT also influenced the results of the color and phenolic compounds of the starch, with a darker appearance due to the temperature to which the starch is subjected during processing, with a decrease in coordinate (L) and an increase in (a* and b*). The AP showed the highest value for gallic acid and quercetin and the AV for proanthocyanidins. Overall, the modification decreased syneresis, thus showing the efficiency in modifying the pigmented rice starch structure and greater stability during storage.

The surface appearance was significantly altered with larger agglomerates, being more visible in the AP_HMT_, where most granules were gelatinized, and the average diameter was greater. Likewise, the type of starch was modified from type A to type Vh, with lower relative crystallinity (15.08%) and lower intensity of peaks in the FT-IR. Checking the textural parameters, the modified red rice starch was significantly altered by the hydrothermal treatment, showing lower values of firmness, gumminess, cohesiveness, and adhesiveness. In general, HMT proved to be a viable and low-cost hydrothermal treatment to modify the analyzed parameters due to changes in the amylose and amylopectin chains, which influence the texture and physicochemical properties of pigmented rice starch. This modification aims to expand the range of applications of rice starch and present greater stability during storage when subjected to temperatures above 100 °C.

## Figures and Tables

**Figure 1 foods-12-04222-f001:**
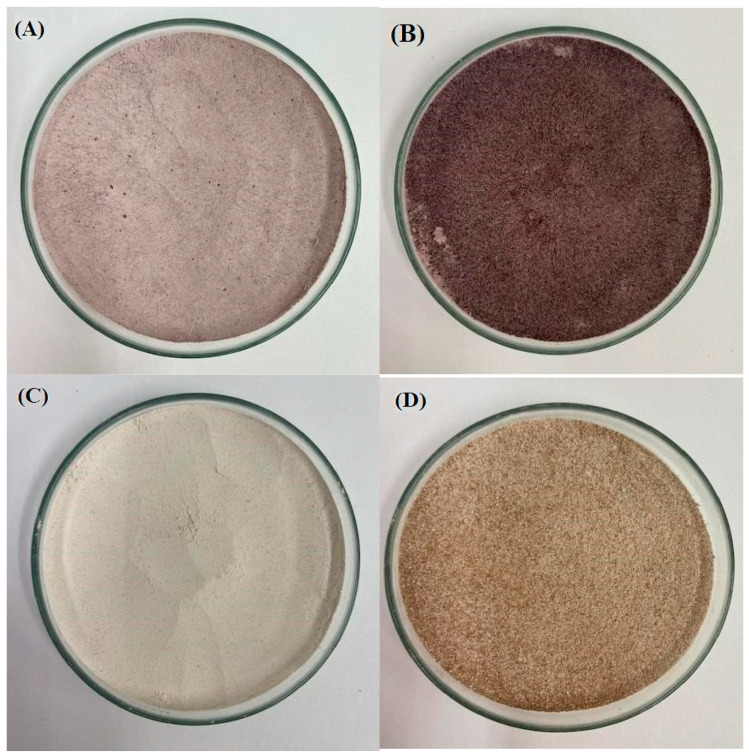
Color modification of pigmented rice starches before and after application of heat–moisture (HMT) heat treatment: (**A**) native black rice starch (AP), (**B**) HMT-modified black rice starch (AP_HMT_), (**C**) native red rice starch (AV), and (**D**) HMT-modified red rice starch (AV_HMT_).

**Figure 2 foods-12-04222-f002:**
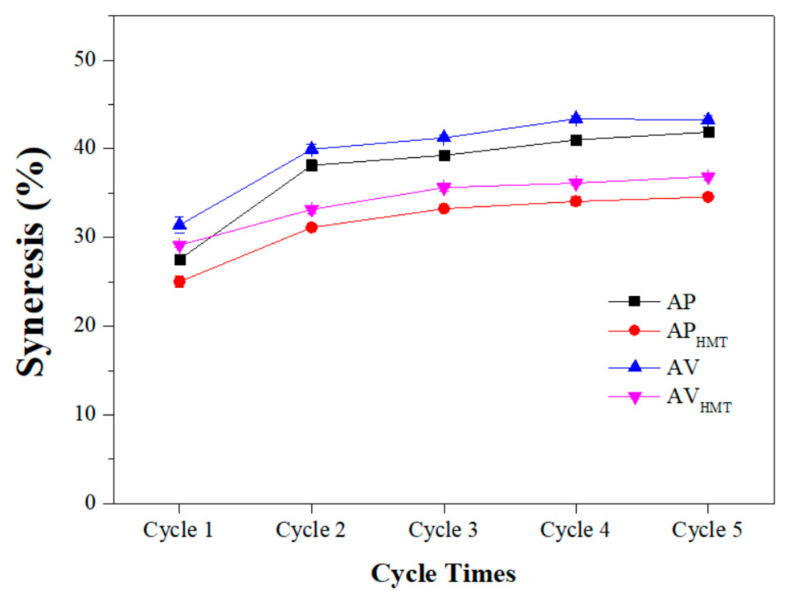
Syneresis index of black and red rice starches, native and modified. AP—Native black rice starch; AP_HMT_—Modified black rice starch; AV—Red rice native starch; AV_HMT_—Modified red rice starch.

**Figure 3 foods-12-04222-f003:**
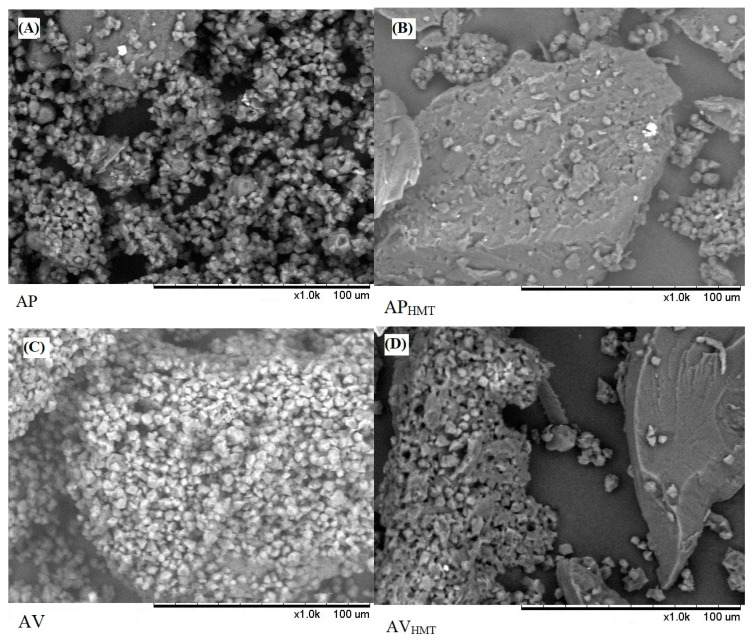
Micrograph of the starch structure of black and red rice, native and modified. (**A**): AP—Native black rice starch; (**B**): AP_HMT_—Modified black rice starch; (**C**): AV—Native red rice starch; (**D**): AV_HMT_—Modified red rice starch.

**Figure 4 foods-12-04222-f004:**
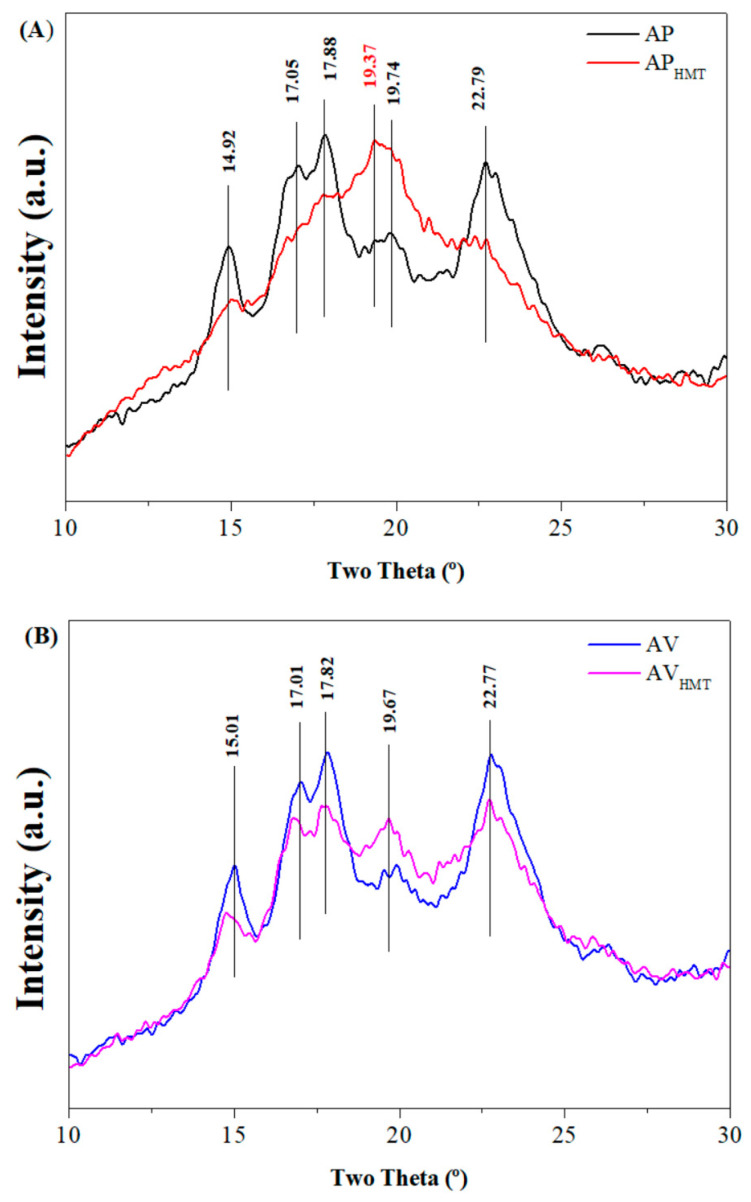
X-ray diffraction (XRD) of starches. (**A**): AP—Native black rice starch and AP_HMT_—Modified black rice starch; (**B**): AV—Red rice native starch and AV_HMT_—Modified red rice starch; (**C**) The interaction of all formulations.

**Figure 5 foods-12-04222-f005:**
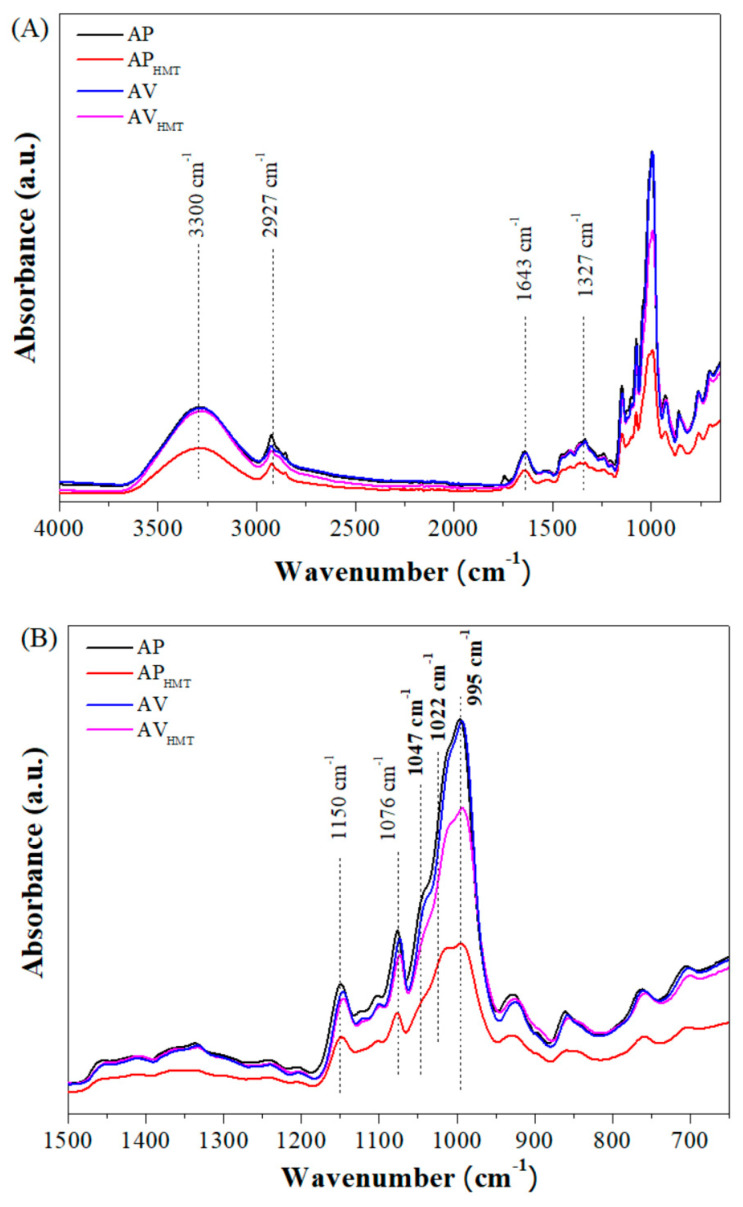
FT-IR of pigmented rice starches. AP—Native black rice starch; AP_HMT_—Modified black rice starch; AV—Red rice native starch; AV_HMT_—Modified red rice starch; (**A**) Total FTIR spectrogram and (**B**) Amplified FTIR spectrogram–wavenumber between 1500–650 cm^−1^.

**Table 1 foods-12-04222-t001:** Evaluation of extraction yield, starch and amylose content, and starch color of black and red rice.

Formulations	Extraction Yield (%)	Starch (g 100 g^−1^)	Amylose (g 100 g^−1^)	Color
AP	52.06 ± 1.14 ^A^	82.06 ± 0.54 ^A^	12.11 ± 0.55 ^B^	(L) 62.71 ± 5.28 ^B^ (a) 4.92 ± 0.08 ^B^ (b) 6.52 ± 0.12 ^C^
AP_HMT_	44.31 ± 1.21 ^B^	73.05 ± 0.62 ^B^	9.39 ± 0.28 ^D^	(L) 28.11 ± 2.14 ^D^ (a) 10.04 ± 1.03 ^A^ (b) 6.84 ± 0.91 ^C^
AV	49.11 ± 0.98 ^C^	81.12 ± 1.06 ^A^	15.46 ± 0.59 ^A^	(L) 71.16 ± 2.62 ^A^ (a) −0.78 ± 0.40 ^C^ (b) 10.16 ± 0.15 ^B^
AV_HMT_	41.07 ± 1.31 ^D^	69.65 ± 0.86 ^C^	10.87 ± 0.67 ^C^	(L) 52.88 ± 9.33 ^C^ (a) 4.64 ± 0.65 ^B^ (b) 20.00 ± 1.19 ^A^

Note: Equal superscript capital letters in the same column indicate that the values did not differ significantly according to Tukey’s test (*p* > 0.05). AP—Native black rice starch; AP_HMT_—Modified black rice starch; AV—Red rice native starch; AV_HMT_—Modified red rice starch; L (lightness), a (redness), and b (yellowness).

**Table 2 foods-12-04222-t002:** Phenolic compounds determined by HPLC for pigmented rice starches (black and red).

Formulation	Gallic Acid (mg of Gallic Acid L^−1^)	Proanthocyanidins (mg of Catechin L^−1^)	Quercetin (mg of Quercetin L^1^)
AP	2.49 ± 0.19 ^A^	0.71 ± 0.06 ^C^	2.96 ± 0.12 ^A^
AP_HMT_	1.52 ± 0.17 ^B^	0.48 ± 0.04 ^D^	2.58 ± 0.09 ^B^
AV	0.91 ± 0.09 ^C^	1.37 ± 0.10 ^A^	1.95 ± 0.18 ^C^
AV_HMT_	0.63 ± 0.11 ^D^	1.02 ± 0.05 ^B^	1.20 ± 0.11 ^D^

Note: Equal superscript capital letters in the same column indicate that the values did not differ significantly according to Tukey’s test (*p* > 0.05). AP—Native black rice starch; AP_HMT_—Modified black rice starch; AV—Native red rice starch; AV_HMT_—Modified red rice starch.

**Table 3 foods-12-04222-t003:** Functional characteristics of black and red rice starches, native and heat-moisture-modified.

Formulation	Water Absorption (g 100 g^−1^)	Oil Absorption (g 100 g^−1^)	Lactose-Free Milk Absorption (g 100 g^−1^)	Whole Milk Absorption (g 100 g^−1^)
AP	63.66 ± 0.11 ^C^	63.55 ± 0.35 ^C^	65.24 ± 0.31 ^D^	64.31 ± 0.26 ^D^
AP_HMT_	73.53 ± 0.26 ^A^	72.12 ± 0.28 ^A^	73.58 ± 0.16 ^B^	73.93 ± 0.11 ^A^
AV	70.01 ± 0.18 ^B^	58.52 ± 0.11 ^D^	70.57 ± 0.09 ^C^	70.40 ± 0.32 ^C^
AV_HMT_	74.01 ± 0.29 ^A^	65.96 ± 0.26 ^B^	74.24 ± 0.27 ^A^	71.36 ± 0.19 ^B^

Note: Equal superscript capital letters in the same column indicate that the values did not differ significantly according to Tukey’s test (*p* > 0.05). AP—Native starch of black rice; AP_HMT_—Modified black rice starch; AV—Native red rice starch; AV_HMT_—Modified red rice starch.

**Table 4 foods-12-04222-t004:** Determination of microstructural parameters of black and red rice starch.

Formulation	Mean Diameter (µm)	Relative Crystallinity (%)	IR (1022/1047) cm^−1^	IR (995/1022) cm^−1^
AP	1.81- 6.12	24.03 ± 0.13 ^A^	1.39 ± 0.02 ^B^	1.28 ± 0.01 ^B^
AP_HMT_	57.63–588.23	15.08 ± 0.39 ^C^	1.34 ± 0.01 ^C^	1.08 ± 0.03 ^C^
AV	1.55–6.28	23.88 ± 0.19 ^A^	1.45 ± 0.03 ^A^	1.32 ± 0.01 ^A^
AV_HMT_	29.41–352.94	21.69 ± 0.32 ^B^	1.33 ± 0.02 ^C^	1.28 ± 0.01 ^B^

Note: Equal superscript capital letters in the same column indicate that the values did not differ significantly according to Tukey’s test (*p* > 0.05). AP—Native black rice starch; AP_HMT_—Modified black rice starch; AV—Native red rice starch; AV_HMT_—Modified red rice starch.

**Table 5 foods-12-04222-t005:** Texture of native and modified black and red rice starch samples.

Formulation	Parameter			
	Firmness (N)	Gumminess (N)	Cohesiveness	Adhesiveness (N.m)
AP	0.52 ± 0.02 ^A^	0.42 ± 0.02 ^A^	0.74 ± 0.01 ^A^	0.49 ± 0.02 ^A^
AP_HMT_	0.46 ± 0.02 ^B^	0.37 ± 0.02 ^B^	0.68 ± 0.02 ^B^	0.47 ± 0.02 ^A^
AV	0.41 ± 0.02 ^C^	0.30 ± 0.02 ^C^	0.63 ± 0.01 ^C^	0.41 ± 0.02 ^B^
AV_HMT_	0.34 ± 0.01 ^D^	0.22 ± 0.02 ^D^	0.52 ± 0.03 ^D^	0.35 ± 0.01 ^C^

Note: Equal superscript capital letters in the same column indicate that the values did not differ significantly according to Tukey’s test (*p* > 0.05). AP—Native black rice starch; AP_HMT_—Modified black rice starch; AV—Red rice native starch; AV_HMT_—Modified red rice starch.

## Data Availability

The data used to support the findings of this study can be made available by the corresponding author upon request.

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
