# Peer review of "Characterization and Evaluation of Heat–Moisture-Modified Black and Red Rice Starch: Physicochemical, Microstructural, and Functional Properties"

_foods, 2023, doi:10.3390/foods12234222_

Round 1

Reviewer 1 Report

1. Abstract: What is the full name of HMT? It should be noted when an abbreviation name shows for the first time.

2. Keywords: Hydrothermal treatment or heat-moisture treatment?

3. 2.2 What’s the method to dry the extracted starch?

4. Figures: The resolution of the figures are not enough to demonstrate the details. Please update all the figures with high resolution (>300 dpi)

5. Is there any change in the shelf life after HMT treatment?

6. More recent references can be referred. Here are some examples:

  1. https://doi.org/10.1016/j.fochx.2023.100785

  2. https://doi.org/10.1016/j.foodchem.2023.136237

  3. https://doi.org/10.1080/10408398.2023.2230287

Author Response

Reviewer 1

Abstract: What is the full name of HMT? It should be noted when an abbreviation name shows for the first time. 

A: We thank the reviewers for their comment. Check the corrections in line 21.

2. Keywords: Hydrothermal treatment or heat-moisture treatment? 

A: Heat-moisture treatment is a type of hydrothermal treatment, as the first word is already in the title, the authors decided to leave hydrothermal treatment in the keywords.

3. 2.2 What’s the method to dry the extracted starch?

A: The method initially described by Bento et al. (2019) and later modified by Almeida et al. (2021b) was used. The extraction method consists of the use of a sodium metabisulfide solution, and unitary operations of crushing, filtration, decantation and drying.

4. Figures: The resolution of the figures are not enough to demonstrate the details. Please update all the figures with high resolution (>300 dpi)

A: The quality of the figures was significantly improved, which is why they were replaced by others with better resolution.

5. Is there any change in the shelf life after HMT treatment? (em “respostas”)

A: Evaluating the starch in powder form, we believe that due to its low water content (below 12%), in accordance with current Brazilian legislation, both formulations have the same durability.

Evaluating gelatinized starch, it is possible to verify by the syneresis index that they present differences when applied to a product in which the starch is subjected to gelatinization temperature and in the presence of water. The relationship we noticed is that modified starches absorb more water, which is why the syneresis index is lower, which means the product's shelf life is longer. In the case of native starch, the shelf life is shorter, as there is no absorption of water or liquid in the product, causing microbiological development when not stored under appropriate conditions.

6. More recent references can be referred. Here are some examples:

1. https://doi.org/10.1016/j.fochx.2023.100785

2. https://doi.org/10.1016/j.foodchem.2023.136237

3. https://doi.org/10.1080/10408398.2023.2230287

A: The articles indicated were used to reinforce the introduction of the article, we thank the reviewer for the recommendation.

Reviewer 2 Report

The manuscript describes the effect of heat-moisture treatment on physicochemical, morphological, and functional properties of colored rice starches. The topic is interesting; however, the manuscript has several problems. In the introduction the authors have stated that the heat-moisture treatment improves the slowly digested starch and resistant starch but have not performed any experiments to determine the digestibility of modified starches. Pasting properties are very critical but the authors have not investigated pasting properties. On the other hand, the results indicate that heat-moisture treatment has negative effects on the physicochemical properties of rice starch. What is the main advantage of heat-moisture treatment for colored rice starches?!

Other comments:

Line 61 introduction: Please write about the shortcomings of native starches and the role of modification, particularly physical modifications in improving the physicochemical properties of starch. You can use the following paper: 10.3390/gels8110693

- The introduction is not cohesive please rewrite it. Lines 66-74 are not at a right place.

- Line 106: Heat-moisture treatment

- Line 157: Determination of phenolic …

-Line 348: Please explain why the color of starches has changed after modification.

- Line 350: Determination of phenolic …

The language should be edited by an English editor.

Author Response

The manuscript describes the effect of heat-moisture treatment on physicochemical, morphological, and functional properties of colored rice starches. The topic is interesting; however, the manuscript has several problems. In the introduction the authors have stated that the heat-moisture treatment improves the slowly digested starch and resistant starch but have not performed any experiments to determine the digestibility of modified starches. Pasting properties are very critical but the authors have not investigated pasting properties. On the other hand, the results indicate that heat-moisture treatment has negative effects on the physicochemical properties of rice starch. What is the main advantage of heat-moisture treatment for colored rice starches?!

A: The part that talks about digestibility was removed from the introduction, we thought this information was important for the reader, since the digestibility of gluten-free products and those with slow digestion of carbohydrates, such as starch, is interesting for diets.

The starch pastes were characterized according to the syneresis index and texture. In addition, the absorption capabilities of water, oil and milk give a good indication of what happens to starch in the formation of pastes with different liquids. Thermal, rheological and viscoamylographic properties were not included in this article. The focus was to analyze both the starch and the dairy dessert compared to the control sample, which is why texture was so important in this article.

Heat-moisture treatment (HMT) offers several key benefits for colored rice starches. It enhances their functional properties, reduces syneresis, improves stability, and enhances clarity in various food and industrial applications. These modified starches are versatile, cost-effective, and can align with clean label trends. Specific advantages depend on the application and customization of the treatment process.

Other comments:

Line 61 introduction: Please write about the shortcomings of native starches and the role of modification, particularly physical modifications in improving the physicochemical properties of starch. You can use the following paper: 10.3390/gels8110693

A: This polysaccharide has limited functionality in its native form, which can affect its applications (Almeida et al., 2022b). Generally, starch chain structures can be altered through biological, thermal, non-thermal, hydrothermal, and chemical methods to achieve desired technological, functional, and nutritional properties (Punia, 2020). Physical methods are more practical, safe, and environmentally friendly for starch modification, where the purpose of physical modification is to alter the internal struc-ture through the gelatinization of native starch granules. The resulting structural changes in starch due to heat-induced gelatinization have stabilizing effects, including improved solubility, stability under different conditions, viscosity and texture control, resistance to retrogradation, transparency, thermal stability, and reduction in syneresis (Xu et al., 2020; Tarahi et al., 2022).

- The introduction is not cohesive please rewrite it. Lines 66-74 are not at a right place.

A: This sentence was removed from the introduction without compromising the understanding of the article.

- Line 106: Heat-moisture treatment

A: This word has been corrected throughout the article.

- Line 157: Determination of phenolic …

A: The sentence was changed as requested.

-Line 348: Please explain why the color of starches has changed after modification.

A: The change in color in starches after heat-moisture modification can occur due to various reasons, including Maillard reactions, degradation of natural pigments, and alterations in the structure and composition of starch (Sindhu; Devi & Khatkar, 2021).

- Line 350: Determination of phenolic 

A: The sentence was changed as requested.

Reviewer 3 Report

This study Authors wanted to evaluate starch from black and red rice modified by heat-moisture, investigating extraction yield, starch and amylose content, color and phenolic compounds. The manuscript is interesting, but needs a little addition. The abstract should contain more information about the marked parameters. In the introduction, examples of other starches modified by this method and their application possibilities should be described in more detail. The discussion should be improved by comparing the results of research with previous studies on starches of various botanical origins. Charts are generally poorly legible, blurred.

Minor editing of English language required. 

Author Response

This study Authors wanted to evaluate starch from black and red rice modified by heat-moisture, investigating extraction yield, starch and amylose content, color and phenolic compounds. The manuscript is interesting, but needs a little addition. The abstract should contain more information about the marked parameters. In the introduction, examples of other starches modified by this method and their application possibilities should be described in more detail. The discussion should be improved by comparing the results of research with previous studies on starches of various botanical origins. Charts are generally poorly legible, blurred.

A: The summary has a word limitation that makes it difficult to enter more data. The introduction has been changed, adding more recent references and showing other types of HMT-modified starch.

The Figures were inserted with Pohigher quality, we apologize for this

Reviewer 4 Report

line 98  Please clarify the word 'organza'.

line 158  Were these bound phenolics that you measured?  You mentioned before that you removed phenols during extraction of the starch.

line 271  Did you do an ANOVA first?

line296 to 298 and 302 to 305  Do not repeat the same information.

line 307 to 310  Does lipid interfere with amylose analyses?

line 348  I do not see the reference for Castro 2014 in the reference list.

line 417  What do you mean by collage?

line 427 to 429 The reverse of this statement was seen in figure 3.  Please revise sentence.

Figure 3 and any other similar figures, identify which figure is which sample in the legend.

line 491  What do you mean by fibers?  Where did the fiber come from?  I do not see fibers in the picture.

Table 4 IR data  Wat are the units for these values?

line 572 to 573  What are the units for IR values?

line 625 A citation is needed for this statement.

line 631 native black and red rice starches.

line 651  This statement is not true for water and milk, please revise sentence.

Author Response

line 98 Please clarify the word 'organza'.

A: Organza mesh is a fabric that aims to filter starch as it has an opening of 0.3 mm, inside it the residue is retained and the starch is filtered in a liquid form.

line 158 Were these bound phenolics that you measured? You mentioned before that you removed phenols during extraction of the starch.

A: The phenols were removed so as not to interfere with the determination of starch, so these soluble compounds were washed with alcohol (80%). In starch extraction, the only reagent used was sodium metabisulfite.

Using these starches, the total phenolic compounds were determined, and the methodology consists of extraction with acetic acid and ethanol, to be able to be passed through HPLC.

line 271 Did you do an ANOVA first? 

A: The data will be expressed as mean ± standard deviation calculated in triplicate for each analysis, where they will be subjected to ANOVA using the Tukey test (p < 0.05) with data processing using Statistica® v.7 (Statsoft, USA).

line296 to 298 and 302 to 305 Do not repeat the same information.

A: These sentences were removed because the values are already found in Table 1.

line 307 to 310 Does lipid interfere with amylose analyses? 

A: Yes, lipids can interfere with amylose analysis, especially when using certain analytical methods. Amylose analysis typically involves the formation of complexes with iodine, which can be affected by the presence of lipids.

Formation of Iodine Complex: The standard method for amylose analysis involves the formation of a complex between amylose and iodine. This complex has a characteristic blue color, and the intensity of this color is used to quantify the amylose content. Lipids, especially those with high melting points or in a crystalline form, can interfere with the formation of this complex by blocking the interaction sites on the amylose molecules.

Rice starch has a low lipid content, this generally shows the purity in the extraction process, when in its native form it obtained levels of 0.26% for black rice starch and 0.14%.

line 348 I do not see the reference for Castro 2014 in the reference list.

A: The reference was updated to (Amorim et al., 2021) and added as requested.

Amorim, I. S., Almeida, M. C. S., Chaves, R. P. F., & Chiste, R. C. (2022). Technological applications and color stability of carotenoids extracted from selected Amazonian fruits. Food Science and Technology, 42, e01922. Doi: /10.1590/fst.01922.

line 417 What do you mean by collage?

A: The term was removed as it was meaningless in the sentence.

line 427 to 429 The reverse of this statement was seen in figure 3. Please revise sentence.

A: The sentence was revised.

Figure 3 and any other similar figures, identify which figure is which sample in the legend.

A: It was verified that only in Figure 1 the presentation of the formulations was not included in the legend.

line 491 What do you mean by fibers? Where did the fiber come from? I do not see fibers in the picture.

A: This information was actually misinterpreted. That's why it was removed from the article.

Table 4 IR data What are the units for these values?

A: The unit of this result is cm-1, this has already been modified throughout the text and in Table 4.

line 572 to 573 What are the units for IR values?

A: The unit of this result is cm-1, this has already been modified throughout the text and in Table 4.

line 625 A citation is needed for this statement.

A: 72. Nishinari, K., Turcanu, M., Nakauma, M., & Fang, Y. (2019). Role of fluid cohesiveness in safe swallowing. npj Science of Food, 3(1), 5. Doi: 10.1038/s41538-019-0038-8.

line 631 native black and red rice starches.

A: It was corrected as requested.

line 651 This statement is not true for water and milk, please revise sentence.

A: This information is actually not correct, which is why it was removed from the conclusion.

Round 2

Reviewer 1 Report

This study sought to evaluate starch from black and red rice modified by heat-moisture, inves-tigating extraction yield, starch and amylose content, color and phenolic compounds. Water and oil absorption capacity, whole milk and zero lactose absorption index, syneresis index and texture were also analyzed. Morphostructural analysis included Fourier transform infrared spectroscopy, X-ray diffraction and scanning electron microscopy. The HMT treatment reduced the extraction yield and the starch and amylose content, with native black rice starch having the highest values for these parameters. The heat-moisture treatment (HMT) promotes physicochemical modifications.

The research is good design and well organized. However, there still have some issues need to check.

1.      The heat-moisture treatment (HMT) promotes physicochemical modifications. It is necessary to give the information about resistant starch, please refer this reference (Comprehensive Reviews in Food Science and Safety, 2023, Doi: 10.1111/1541-4337.13217).

2.      Line 59-60. “Black rice stands out among pigmented rice varieties because it has good sensory characteristics and high nutritional value”. In fact, there are some polyphenols in the wholegrain (Food & Function, 2022, 13(24), 12686-12696).

3.      Line 80-81. “The most recent applications of HMT-modified starches are for the quality of 3D printed wheat starch gels”. It can be bioactive delivery system (Critical Reviews in Food Science and Nutrition, 2023, Doi: 10.1080/10408398.2023.2230287).

4.      “2.7 Determination of phenolic compounds by High-Performance Liquid Chromatography (HPLC)”. It is better to refer the reference for gradient elution about phenolic acid (International Journal of Food Science and Technology, 2020, 55(6): 2531-2540).

5.      Line 306-307. It should have standard error for M±SD.

6.      It is better to supplement HPLC spectrogram.

7.      The expression should be verified in the manuscript.

8.      The reference should be updated in recent years.

This study sought to evaluate starch from black and red rice modified by heat-moisture, inves-tigating extraction yield, starch and amylose content, color and phenolic compounds. Water and oil absorption capacity, whole milk and zero lactose absorption index, syneresis index and texture were also analyzed. Morphostructural analysis included Fourier transform infrared spectroscopy, X-ray diffraction and scanning electron microscopy. The HMT treatment reduced the extraction yield and the starch and amylose content, with native black rice starch having the highest values for these parameters. The heat-moisture treatment (HMT) promotes physicochemical modifications.

The research is good design and well organized. However, there still have some issues need to check.

1.      The heat-moisture treatment (HMT) promotes physicochemical modifications. It is necessary to give the information about resistant starch, please refer this reference (Comprehensive Reviews in Food Science and Safety, 2023, Doi: 10.1111/1541-4337.13217).

2.      Line 59-60. “Black rice stands out among pigmented rice varieties because it has good sensory characteristics and high nutritional value”. In fact, there are some polyphenols in the wholegrain (Food & Function, 2022, 13(24), 12686-12696).

3.      Line 80-81. “The most recent applications of HMT-modified starches are for the quality of 3D printed wheat starch gels”. It can be bioactive delivery system (Critical Reviews in Food Science and Nutrition, 2023, Doi: 10.1080/10408398.2023.2230287).

4.      “2.7 Determination of phenolic compounds by High-Performance Liquid Chromatography (HPLC)”. It is better to refer the reference for gradient elution about phenolic acid (International Journal of Food Science and Technology, 2020, 55(6): 2531-2540).

5.      Line 306-307. It should have standard error for M±SD.

6.      It is better to supplement HPLC spectrogram.

7.      The expression should be verified in the manuscript.

8.      The reference should be updated in recent years.

Author Response

Dear reviewer,

Firstly, we would like to thank you for the opportunity to improve our manuscript.

This study sought to evaluate starch from black and red rice modified by heat-moisture, investigating extraction yield, starch and amylose content, color and phenolic compounds. Water and oil absorption capacity, whole milk and zero lactose absorption index, syneresis index and texture were also analyzed. Morphostructural analysis included Fourier transform infrared spectroscopy, X-ray diffraction and scanning electron microscopy. The HMT treatment reduced the extraction yield and the starch and amylose content, with native black rice starch having the highest values for these parameters. The heat-moisture treatment (HMT) promotes physicochemical modifications.

The research is good design and well organized. However, there still have some issues need to check.

  1. The heat-moisture treatment (HMT) promotes physicochemical modifications. It is necessary to give the information about resistant starch, please refer this reference (Comprehensive Reviews in Food Science and Safety, 2023, Doi: 10.1111/1541-4337.13217).

A: HMT can also lead to the formation of resistant starch (undigested in the small intes-tine) upon cooling, where part of the starch recrystallizes into a more organized form known as resistant starch type 3 (RS3), which is beneficial for promoting intestinal health and aiding in blood glucose level regulation (Wu et al., 2023).

  1. Line 59-60. “Black rice stands out among pigmented rice varieties because it has good sensory characteristics and high nutritional value”. In fact, there are some polyphenols in the wholegrain (Food & Function, 2022, 13(24), 12686-12696).

A: HMT can also lead to the formation of resistant starch (undigested in the small intes-tine) upon cooling, where part of the starch recrystallizes into a more organized form known as resistant starch type 3 (RS3), which is beneficial for promoting intestinal health and aiding in blood glucose level regulation (Wu et al., 2023).

  1. Line 80-81. “The most recent applications of HMT-modified starches are for the quality of 3D printed wheat starch gels”. It can be bioactive delivery system (Critical Reviews in Food Science and Nutrition, 2023, Doi: 10.1080/10408398.2023.2230287).

A: This reference was inserted in the introduction as a review of the application of HMT-modified starches. We thank the reviewer for the suggestion.

  1. “2.7 Determination of phenolic compounds by High-Performance Liquid Chromatography (HPLC)”. It is better to refer the reference for gradient elution about phenolic acid (International Journal of Food Science and Technology, 2020, 55(6): 2531-2540).

A: Next, identification was carried out under the same treatment conditions as the respective standards, and the quantification of the phenolic compound was based on re-tention time (GA: 3.41 min; P: 10.4 min and Q: 12.75 min) and peak area.

This methodology was based on the reference by Almeida et al. (2021a) for red rice starch.

  1. Line 306-307. It should have standard error for M±SD.

A: The data will be expressed as mean ± standard error (M ± SD) calculated in tripli-cate for each analysis, where they will be subjected to ANOVA using the Tukey test (p < 0.05) with data processing using Statistica® v.7 (Statsoft, USA).

  1. It is better to supplement HPLC spectrogram.

A: The authors did not understand which expression should be modified in the text, they failed to identify where in the text.

  1. The expression should be verified in the manuscript.

A: As the authors preferred to adopt the quantitative method to have the power to compare with other samples, it was decided that the values would be calculated using the data given by HPLC, in triplicate and the deviation made. Only in this way was it possible to pass the ANOVA and the Tukey test. The authors respect the reviewer's comment, but in starch articles the spectrograms are not included as the information would be repeated to that contained in the table. As we work with 4 samples, if we were to add them, there would be 12 graphs, without numerical value, just showing the peaks that were found and justifying them by the value in min. Therefore, the value in min of each peak found was added to the methodology.

  1. The reference should be updated in recent years.

A: Older references were removed and new ones were added in their place.

Reviewer 2 Report

The manuscript is acceptable.

Author Response

The authors would like to thank you for the opportunity to improve our manuscript with the review!

Reviewer 3 Report

Manuscrypt has been revised according to Reviewers' comments/suggestions.

Author Response

(The authors gave the same response as above.)
